# Application of Stray Magnetic Field for Monitoring the Wear Degree in Steel Components of the Lift Guide Rail System

**Poul Lonkwic** [1,*]📷, **Tomasz Krakowski** [2] **and Hubert Ruta** [2]

[1] The State School of Higher Education, The Institute of Technical Sciences and Aviation, Pocztowa Street 54, 22-100 Chełm, Poland

[2] AGH University of Science and Technology, Faculty of Mechanical Engineering and Robotics, Adama Mickiewicza Ave. 30 (B2-p115), 30-059 Kraków, Poland; krakowsk@agh.edu.pl (T.K.); hubert.ruta@agh.edu.pl (H.R.)

\* Correspondence: plonkwic@gmail.com; Tel.: +48-884-883-464

**Abstract:** This paper presents the results of forced wear simulation of the friction lift guide rails. The forced wear in the case discussed is an effect of plastic strain of the guide rail surface due to emergency braking of the lift. For the purpose of qualitative and quantitative assessment of wear, the authors applied the numerical simulation of a stray magnetic field. Application of this method allowed evaluating the degree of wear based on the stray field changes. Application of this simulation method allowed obtaining satisfactory results of qualitative and quantitative assessment of the guide rail wear. The intention of this paper was to prove that the permanent magnetic field and the stray field can be applied for the efficient detection of the steel guide rail damages and to verify the possibility of making the quantitative assessment related to the guide rail wear degree versus the personal lift service life.

**Keywords:** wear; wear of guide rails; elevator; magnetic field; finite element method (FEM); monitoring; service life; steel

---

## 1. Introduction

Monitoring the technical condition of facilities is broadly described in the worldwide literature. This process aims not only at evaluating their further operation but also at increasing safety and preventing unscheduled downtime, which directly affects the economic result.

Currently, all technical facilities are subject to monitoring and the older ones used in industry are being adapted to the changing regulations. The monitored parameters are indicated in Figure 1.

In publication [1], the author presented his own invented system for monitoring the condition of the bonded joints in belt conveyors used in the mining industry. The author indicated weak and strong points of such systems and described his own concept of the system. The proposed monitoring system of the author's own invention could be used, among others, in the mining sector. Another approach to the wire rope monitoring process in real time is the method described in publication [2]. The authors proposed the application of visual systems that can be used for the lifting wire rope quality and wear degree analysis through measurement of their diameters and significant parameters of the wire rope lay. These systems enable detecting the surface damages and their type with an accuracy that is much higher than those of other methods being applied, which in turn is related to the increased level of their safety of use. In the conference materials [3], the authors presented the metal magnetic memory (MMM) method for monitoring the technical condition of a hoisting machine shaft in real time. Application of this method enabled the authors to passively observe the magnetic field changes

and the emerging magneto-mechanical effects for contactless monitoring of the critical components of the mining machines. Based on an example of the hoisting machine shaft, the authors discussed the diagnostic problem and gave a theoretical background of magneto-mechanical effects, thus describing the impact of the electrical motor magnetic field, metal magnetic memory (MMM) method, and the expected diagnostic symptoms. In publication [4], the author described the implementation of a new and innovative solution for the diagnostic system using artificial intelligence to continuously assess the technical condition of a conveyor belt with steel ropes. Based on the results of tests carried out on the real object in the coal mines, Marcel and Jankowice defined how information technology can be used to reduce operational costs by increasing the durability and reliability of conveyors used in mining. In the author's opinion, the developed system of monitoring has an important role for the conveyor operation quality. Based on the analysis of data recorded in the measuring system damage matrix, the affiliation functions are defined for use in the expert system. The monitoring system presented in this article enables increasing the durability and reliability of the continuous coal transport systems using artificial intelligence. In publication [5], the author presented the possibilities of assessing the technical condition (quality analysis) of compact wire ropes by applying wavelet analysis, which is an important element supporting the decision process during non-destructive testing. In the author's opinion, this was the first information related to the issues of diagnosing the new design of wire ropes. This publication presented the analysis results of the signal recorded on the new compact wire rope with the damages being modeled. The choice of wavelet type and the choice of decomposition detail carrying the information about the wear symptoms as well as the choice of recording frequency were very important elements of the diagnosing process. The tested wire rope surface replicas were very helpful for technical condition assessment. In article [6], the authors focused their attention on the issues related to monitoring the lifting wire ropes by means of magnetometric sensors intended to determine the relation between the number of bends of the steel rope and the value of its magnetic field induction. The authors related the obtained results to the wire ropes operating on real objects, which enabled determining their stress and condition. The methodology developed by the authors enabled stating the changing values of magnetic induction in the wire rope being bent after each subsequent bend, which in turn enabled evaluating the number of wire rope bends in relation to the service life. In article [7], the authors attempted to diagnose the condition of the gearbox and indicate the possible causes of its faulty operation. The obtained results were recorded for a passenger lift with a lifting capacity of 630 kg. The signals being subject to monitoring and analysis were noise volume, acceleration, vibration frequency, and heat-affected zones. In publication [8], the authors made an attempt to assess the comfort of the lift passengers upon monitoring the following parameters in various operational conditions: acceleration, acoustic pressure level, and acoustic power in the passenger lift cabin. The cabins from different installation periods and with different wear conditions were assessed. The assessment consisted of data analysis obtained from the accelerometer located in the central part of the cabin and the acoustic parameters obtained by the use of a sound analyser fitted out with a probe with a double microphone. The authors attempted to determine the impact of the technical condition of the frame-to-cabin system on the values of the recorded parameters. The authors endeavored to answer the question of whether there was a correlation between the technical condition of the selected lift components and the level of comfort while riding. In publication [9], the authors discussed the issue of energetic efficiency of personal lifts. They presented the discrepancies resulting from various standards of computations concerning the energetic effectiveness of such hoisting and hauling equipment, emerging mainly from the accepted variety of algorithms and subjective assumptions that should be used for making the computations. A significant problem that they pointed out referred to theoretical nature of such computations being in isolation from the real operational loading of the object. They presented their own equipment solutions that enabled measuring the actual operational and energetic parameters of lifts, and at the same time, they proposed their own method procedure for the determination (computation) of the energetic effectiveness of the lifts.

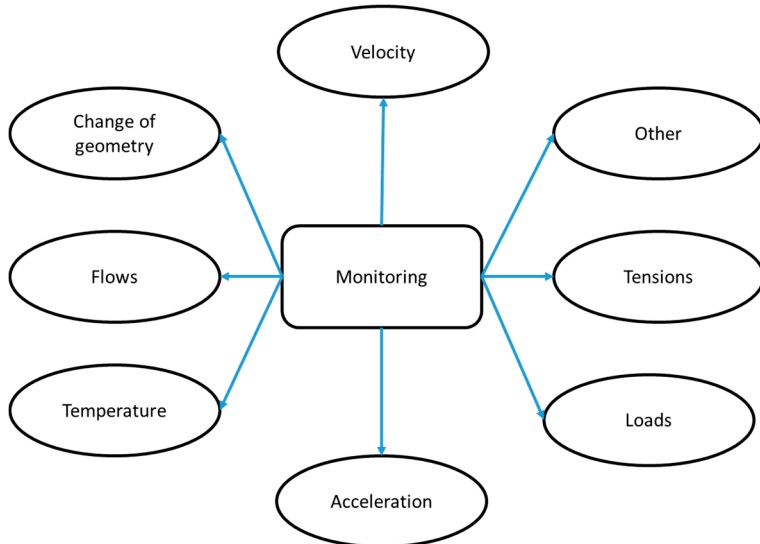

**Figure 1.** Monitoring technical facilities.

Another big area that the scientists were interested in was the monitoring of issues related to the material removal processing. In publications [10,11], the authors observed and influenced the process parameters while drilling, milling, or rolling using, for this purpose, the signals received by monitoring. In publication [10], the authors used the Huang–Hilbert transform to assess the signals received while milling with the end mill of special "Hi-Feed" shape. For the purpose of monitoring the processing condition, the authors used the vibroacoustic sensors that recorded the tool-object system vibration. The obtained signal analysis results proved that there are amplitudes and frequencies of empirical modal constituents that allow separating various vibration elements caused by phenomena related to the drive system and the machine components. Based on the signals obtained, the authors claim that there are circumstances that enable assessing the correct performance of the milling process, the selection of milling parameters, and the milling cutter wear degree. In the conference materials [11], the authors presented the results of designing and monitoring the drilling process. For this purpose, they applied the vibroacoustic sensors and used the obtained spindle vibration signals for the Huang–Fourier transform. During the tests, the authors simulated the drilling process having various processing parameters, and on that basis, they classified the signals using the Fourier spectra and the signal envelope curves.

In publications [12–14], the authors presented the results of their own mathematical and statistical analysis of results obtained during the emergency braking of a lift, including time-lag values and braking distance having an impact on the durability and life of the lift guide rails.

Summing up the above, it can be stated that the current issues related to the transport of goods and people refer not only to the correct selection of the means of transport to meet the requirements, e.g., that it be failure-free, safe, quick, etc., but also cover the issues related to the selection of appropriate components and systems as well as monitoring and controlling their operation in real time.

## 2. Lifting Machine Service Life

The lifting machine service life definition provided in the Regulation of the Minister of Entrepreneurship and Technology specifies the service life term in the context of the lifting machines. The service life should be understood as the limiting parameters used for the assessment and identification of technical condition, which are defined on the basis of the number of operation cycles and loading condition of the hoisting and hauling equipment within the assumed operation period allowing for the actual operation conditions [15]. One of the elements in the lifting installations is the steel guide rails (Figure 2) constituting the cabin guiding system in shafts and at the same time fulfilling an important role in the process of emergency braking when the safety gears get engaged.

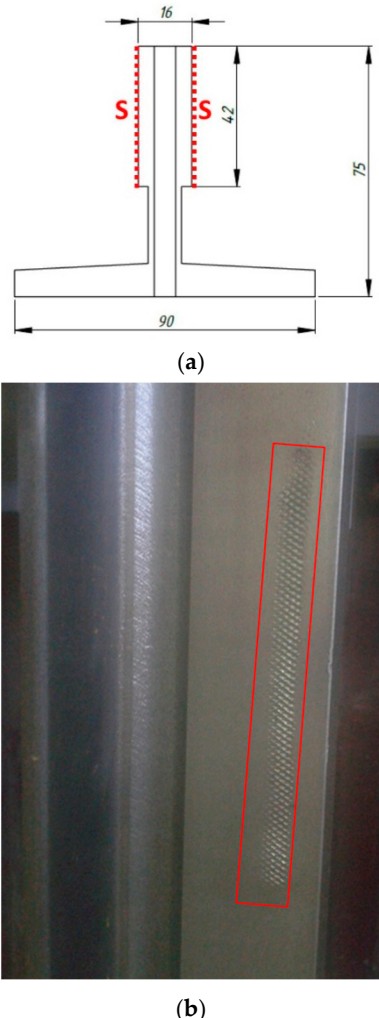

**Figure 2.** Rail T90/A: (**a**) drawing—cross-section, (**b**) view with plastic damage on guide rail working surface.

This phenomenon is a strongly dynamic process that results in wear. When the braking units get engaged during movement of the cabin at excessive velocity, the velocity limiter is activated, the safety gears of the brake are released, and its elements get wedged on the guide rails. The mutual contact of elements and big values of forces that happen to reach more than 20 kN result in plastic strains (Figure 2b) on the track of guide rails (Figure 2a). Prior to further acceptance of the object to operation, the safety gear interference locations need leveling up by grinding and the smoothing of notches that emerged while braking. The guide rails' wear increases due to the long service life of lifting systems and occurrences related to emergency braking during normal operation as well as during acceptance tests performed by relevant offices (in Poland—Office of Technical Inspection). Thus, monitoring the technical condition of lift guidance systems is required in the context of the lift service life.

The guide rails are of steel, have the T-shaped cross-section, and are made by drawing, rolling, or grinding the working surface S (Figure 2a). The mechanical and strength properties of the materials used for the manufacture of the E235B guide rails per ISO 630:1995 are presented in Table 1.

**Table 1.** Mechanical and strength properties of the guide rail material.

| Name of Variable | Value | Unit |
|---|---|---|
| density | 7.8 | g/cm$^3$ |
| Young's modulus | 23% at 20 °C | - |
| Poisson's ratio | 0.29 | - |
| tensile strength R$_m$ | 405 | MPa |
| yield point R$_e$ | 210 | MPa |

In this article, the authors present the concept of using the stray magnetic field measurement for the identification of wear locations and assessment of the lift guide rail wear degree. They were inspired to address this issue due to the market demand as well as due to the innovative solutions developed by Poul Lonkwic and protected by patent no. PL423758, which was met with big approval by the Elevator World International Association of Lift Manufacturers and has been granted its first award as the Project of the Year 2020. The proposed method using the stray field measurements was applied, among others, in the heads for steel rope testing by the magnetic method Magnetic Rope Testing (MTR), which is commonly known and used worldwide. Another application is the magnetic powder method Magnetic particle Inspection (MTI), which is recognizable worldwide and enables performing non-destructive testing (NDT) of ferromagnetic elements in which a stray field is detected by means of magnetic suspended solids.

After appropriate preparation of the surfaces to be tested, the majority of non-destructive tests allow detecting the damages on the guide rails that occur when the lifting machine safety gears are engaged. When the guide rail is clean and free from grease and other types of operational contamination, the damage can be localized even by visual inspection. The undertaken activities were intended to be applied during the diagnostic process; the method can be automated, does not require shutdown of the lifting machine for a long period of time, and provides clear results after a short duration of testing (during the lift cabin traveling between the extreme stops), while the applied equipment does not require any previous calibration depending on the geometrical dimensions of the tested guide rail.

An important functional requirement was also to apply a method that does not require any external power supply source. This shall significantly facilitate testing the guide rail sections with lengths that may reach several dozens of meters. The requirements presented above, the authors' experience in designing the magnetic cores, and application of the MTR method for testing the steel load-carrying cables were the reasons why the authors decided to use the described method for the assessment of technical condition of the lift guide rails.

The solution proposed in the article is based on the application of a permanent magnetic field in the magnetic circuit together with the possible use of magnetic field sensor (for example hall or inducive). In accordance with the major assumptions, the magnetic circuit of the measuring head (Figure 3) consists of two permanent magnets (B) as the sources of the magnetic field and a cramp for the unilateral closing of the circuit (C). The other side of the circuit is closed by the structure of the very guide rails (A) that are subject to testing (monitoring). During measurement, the head is moved along the guide rail path to evaluate the technical condition of the guidance system over its entire length. In the locations with notches or local changes in the cross-section caused by safety gears or subsequent grinding, there is stray magnetic field disturbance in the guide rails' vicinity. Such disturbance consists of a sudden change of the magnetic field induction vector and thus a change of values of particular magnetic field induction components *Bx*, *By*, and *Bz*. The values of the particular components are registered by means of the magnetic field sensor located close to the guide rail track on the opposite side of the magnetization system.

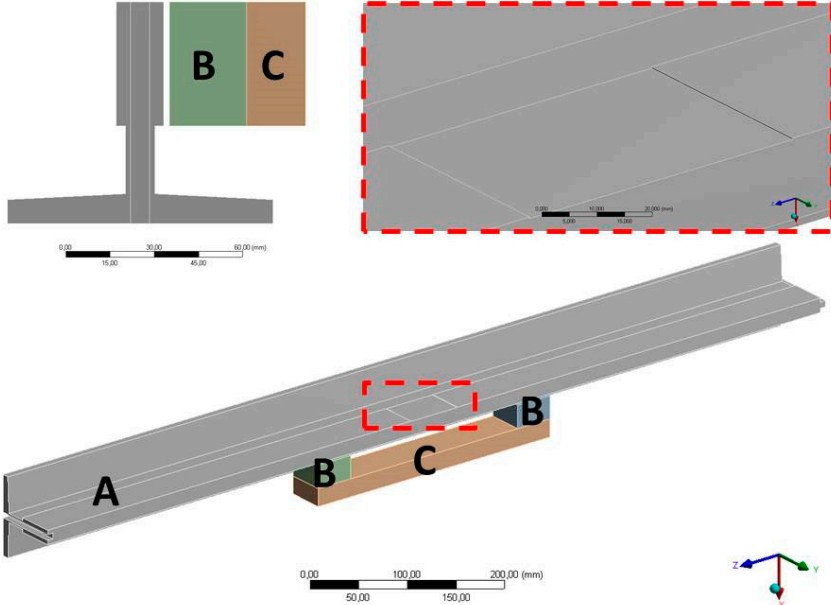

**Figure 3.** Model subject to numerical analysis.

The appropriate design parameters (material dimensions and parameters) for generating the optimum magnetic core were chosen based on experience and the methods applied for generating the magnetic head circuits for testing the steel ropes, the wire ropes at terminations in coned seats, and the wire ropes in the belt conveyors [16,17]. For many years already, the numerical methods have been growing in popularity for prototyping the magnetic circuits in devices of various types. Among the boundary element method, finite difference method, and finite element method, the most appropriate one for electromagnetic issues is the finite element method.

The mathematical and physical apparatus of the finite element method for electromagnetism is based on Maxwell field formulas in a differential form. Due to the development of simulation tools, the manual solving of such formulas has been converted to a form enabling the computation of magnetic field potentials; as a result, writing source codes containing such dependences became redundant, and the computation role was taken over by advanced solvers of numerical systems, such as ANSYS (Release 15, Worldwide Headquarters ANSYS, Inc., Canonsburg, PA, USA). Nonetheless, to carry out the scientific considerations, it is essential to understand this mathematical and physical apparatus and the constitutive formulas that define it. Thanks to this, the use of numerical tools becomes effective and leads to a deliberate interpretation of results.

## 3. Description of Magnetic Field by Differential Method for FEM Purposes

The electromagnetic field distribution in any 3D space is determined by four Maxwell formulas. They can take a differential (as below) or integral form. At the same time, these are the basic formulas for computational systems that make use of FEM or edge element method (MEB), including, among others, the ANSYS system used in this research, where Equations (1) and (3) refer to the magnetic field description, while Equations (2) and (4) refer to the electric field [18–22].

$$\nabla \times \{H\} \;=\; rot\{H\} \;=\; \{J\} + \left\{\frac{\partial D}{\partial t}\right\} \tag{1}$$

$$\nabla \times \{E\} \;=\; rot\{E\} \;=\; -\left\{\frac{\partial B}{\partial t}\right\} \tag{2}$$

$$\nabla \cdot \{B\} \;=\; div\{B\} \;=\; 0 \tag{3}$$

$$\nabla \cdot \{D\} \;=\; div\{D\} \;=\; \rho \tag{4}$$

where

{ } = designation of example vector value

[ ] = designation of example matrix

{H} = magnetic field intensity vector [A/m]

{J} = electric current density vector [A/m$^2$]

{D} = electric field induction vector [C/m$^2$] (Maxwell called it a movement vector, but to avoid misunderstanding and coincidence with mechanical movement, here we use the electric flux density or electric induction term.)

$t$ = time [s]

{E} = electric field intensity vector [V/m]

{B} = magnetic field induction vector [T] or [Wb/m$^2$]

$\rho$ = total density of electric load [C/m$^3$].

Thus, based on the system of Maxwell formulas, we can finally derive the following formulas of continuity use in the ANSYS computation procedure [23]:

$$\nabla \cdot \left[ \{J\} + \left\{ \frac{\partial D}{\partial t} \right\} \right] = 0. \tag{5}$$

In the case of issues referring to the absorptive materials (except for permanent magnets), i.e., magnetically soft materials-ferromagnetic materials, the constitutive relation for the dependence between magnetic field intensity and magnetic induction is expressed in Equation (6) [20].

$$\{B\} = [\mu]\{H\} \tag{6}$$

where

$[\mu]$ = magnetic permeability matrix being generally the magnetic field intensity vector function (the defining methods are specified below).

The matrix $[\mu]$ is defined by the field function (due to the considered nonlinearity of material properties) and has the following form [23]:

$$[\mu] = \mu_h \begin{bmatrix} 1 & 0 & 0 \\ 0 & 1 & 0 \\ 0 & 0 & 1 \end{bmatrix} \tag{7}$$

where

$\mu_h$ = magnetic permeability obtained from the defined input curve B versus function H.

Considering the ANSYS computation procedures, the major constitutive formulas being the derivatives of Maxwell formulas take the form as shown below. In case of the considerations allowing for the use of permanent magnets, as in the mentioned analysis, the constitutive relation takes the following form [23]:

$$\{B\} = [\mu]\{H\} + \mu_0\{M_0\} \tag{8}$$

where

$\{M_0\}$ = residual flux density vector of internal magnetization (vector of magnetization intensity—magnetization)

Considering the reluctivity categories, the general constitutive formula takes the following form [23]:

$$\{H\} = [v]\{B\} - \frac{1}{v_0}[v]\{M_0\} \tag{9}$$

where

$[\nu]$ = reluctivity matrix = $[\mu]^{-1}$
$\nu_0$ = vacuum reluctivities.

Applying the numerical computations, the iteration process allows for nonlinear properties of the phenomena and of the very materials. For each subsequent iteration, a constant value of material parameters $(\varepsilon, \mu, \sigma)$ is accepted in a given space of the considered computational area. In order to take account of the anisotropy phenomena, the scalars of the above-mentioned material parameters are superseded by the tensor quantities [20]. Considering the supplementary constitutive relations concerning material parameters, the Maxwell formulas take the following form:

$$\nabla \times \{H\} = rot\{H\} = \{J_s\} + \{J_s\} + \left\{\frac{\partial D}{\partial t}\right\} = \{J_s\} + [\sigma]\{E\} + [\varepsilon]\left\{\frac{\partial E}{\partial t}\right\} \tag{10}$$

$$\nabla \times \{E\} = rot\{E\} = -\left\{\frac{\partial B}{\partial t}\right\} \tag{11}$$

$$\nabla \cdot \{B\} = div\{B\} = 0 \tag{12}$$

$$\nabla \cdot \{D\} = div\{D\} = div[\varepsilon]\{E\} = \rho. \tag{13}$$

The vectors of five quantities occurring in the Maxwell formulas must meet, at the boundary of two surroundings, the following conditions expressed in the scalar form:

$$E_{1\tau} - E_{2\tau} = 0 \tag{14}$$

$$H_{1\tau} - H_{2\tau} = \eta \tag{15}$$

$$D_{1n} - D_{2n} = \rho \tag{16}$$

$$B_{1n} - B_{2n} = 0 \tag{17}$$

$$J_{1n} - J_{2n} = 0 \tag{18}$$

where

$\eta$ = surface current density at the boundary of two surroundings
$\rho$ = surface load density at the boundary of two surroundings.

The Maxwell formulas are not a direct subject of solution in iterative (numerical) methods of electromagnetic field analysis. Basically, an indirect solution is applied that is based on the definition of a pair of potentials and the relevant solution of boundary or boundary-initial issues. When the distribution of such potentials is known, the vectors of field-characteristic quantities, i.e., $\{E, D, H, B, J_p\}$ [19,24–26] are determined by using the definition relations.

Taking into account only and exclusively the magnetic (magnetostatic) field as in the issue mentioned, we use only two out of four Maxwell formulas for the homogeneous surrounding and reduce it to the following form [19,26]:

$$\nabla \times \{H\} = rot\{H\} = \{J_s\} \tag{19}$$

$$\nabla \cdot \{B\} = div\{B\} = 0. \tag{20}$$

Magnetostatics means that the field change effects in time are ignored. The magnetic field is a vorticity field, so we use the vector magnetic potential defined as [18,20] to describe it:

$$\{B\} = rot\{A\} = \nabla \times \{A\} \tag{21}$$

$$div\{A\} = \nabla \cdot \{A\} = 0. \tag{22}$$

Considering the supplementary constitutive relations between the vectors {B} and {H}:

$$\{B\} = [\mu]\{H\} \tag{23}$$

and substituting the relation (21) defining the vector magnetic potential {A} to the first Maxwell formula for magnetostatics (19), we receive the following partial differential formula describing the magnetic field distribution [18,21]:

$$\nabla \times \left( \frac{1}{[\mu]} \nabla \times \{A\} \right) = \{J_s\}. \tag{24}$$

In case of the homogeneous surrounding, this formula takes the following form:

$$\nabla \times (\nabla \times \{A\}) = [\mu]\{J_s\}. \tag{25}$$

When applying the vector identity appropriate to any vector using the so-called vector Laplace operator (Laplacian), which in the Cartesian coordinate system takes the form perceived as a special case of the Lagrange formula, we receive [18,21]:

$$\nabla \times (\nabla \times \{A\}) = \nabla(\nabla \cdot \{A\}) - \nabla^2\{A\}. \tag{26}$$

Considering the second condition (22) defining the vector magnetic potential {A}, we receive:

$$\nabla^2\{A\} = -[\mu]\{J_s\}. \tag{27}$$

The obtained differential formula, together with the below-mentioned continuity conditions for *H* (28) and *B* (29) at the boundary of any surroundings (e.g., magnetic core-air or pole shoe-cramp) and together with the boundary conditions, enables comprehensively describing the static magnetic field in the area under consideration. Most often, it is used for the field computation in non-homogeneous surroundings [18,20]:

$$H_{1\tau} - H_{2\tau} = \eta \tag{28}$$

$$B_{1n} - B_{2n} = 0. \tag{29}$$

In the case of numerical analysis of a magnetic field in the Cartesian 3D space, the issue described with formula (27) is reduced to the solution of three differential formulas having the same form as the above-mentioned formula for particular components of the vector $\{A\} = \begin{bmatrix} A_x & A_y & A_z \end{bmatrix}$ and the vector $\{J_s\} = \begin{bmatrix} J_{sx} & J_{sy} & J_{sz} \end{bmatrix}$. This requires making very complicated numerical computations and generates difficulties in the determination of boundary conditions of particular components of the vector {A}. The situation gets still more complicated regarding computations when the considerations are made in the cylindrical or spherical coordinate system. This is related to the difficulties in separating the formulas into the partial differential formula system for particular three components of the {A} vector potential. For this reason, the concept of magnetic potential scalar [19] is introduced to the computations. Let us assume that the magnetic field intensity vector in the linear surrounding can be recorded in the form of two components [26].

$$\{H\} = \{H_s\} + \{H_m\} \tag{30}$$

The first component $\{H_s\}$ corresponds to the magnetic field intensity vector in the homogenous surrounding whose source is the current, and we can determine it using the Biot-Savart law. The second component is the magnetic field intensity vector resulting from magnetization of the material

surrounding located in the field interaction zone. Regarding the mentioned components, we can record the relation [18,20]:

$$rot\{H_s\} \;=\; \{J_s\} \tag{31}$$

$$rot\{H_m\} \;=\; 0. \tag{32}$$

It turns out from Equation (31) describing the rotation of vector $\{H_m\}$ and from the differential identity $rot(grad\phi) \equiv 0$ that the magnetic field intensity $H_m$ may be determined by the scalar function gradient. This function is identified as $\phi$ and is called the scalar magnetic field potential [18,20]:

$$\{H_m\} \;=\; -grad\phi. \tag{33}$$

By substituting Equation (30) and the formula of scalar magnetic potential (33) to the second Maxwell formula for magnetostatics (20), we derive the formula that enables determining the value of scalar magnetic potential $\phi$ at any point [18,20]:

$$div([\mu]\{H_s\}) \;=\; div([\mu]grad\phi). \tag{34}$$

Considering the above, the magnetic induction vector at any point shall be expressed by the following relation:

$$\{B\} \;=\; [\mu]\{H\} \;=\; [\mu](\{H_s\} - grad\phi). \tag{35}$$

Equations (34) and (35) enable computing the potential and the magnetic field induction at the nodes of the discretized area elements being subject to the finite element method analysis (Figure 6). Then, using the functions of the $B$ value shape, they are approximated to the inter-node areas. As a result of the mentioned computations, we receive the response related to the $B$ magnetic induction within the entire volume in the space analyzed (guide rail, head, air surrounding) (Figures 8 and 9).

## 4. Geometrical Model

The target geometrical model submitted to final numerical analysis consists of 1 running meter of the guide rail of T90/A type having the transverse section, as shown in Figure 2 (Figure 3—"A" mark). The permanent magnets (B) having the square section, dimensions of $42 \times 42$ mm, and the height of 26 mm, fulfilling also the function of pole shoes, form the 3-mm air gaps between the guide rail and the magnet pole plane. The outer spacing of the magnets is 300 mm. The circuit is closed by the magnetic cramp (C) of the width determined by the size of the applied magnets and the thickness of 20 mm. On the track of a guide rail, on the opposite side of the magnetic circuit, a defect having the initial width $sz = 60$ mm and the depth $g = 0.3$ mm was modeled. One of the analysis assumptions was to verify the possibility of qualitative and quantitative assessment of the guide rail wear degree.

For this reason, the impact of the defect size onto the stray field nature and parameters was analyzed for the assumed variability range defined for $sz = 50 \div 150$ mm and $g = 0.3 \div 2.0$ mm. The obtained effects are presented in the further part of the article. One of the requirements for the performance of numerical magnetostatic analyses is to define the air space of the surrounding around the magnetic core and the guide rail where, among others, the stray field develops. In the mentioned case, such space was defined as an air cuboid with the boundary areas separated from the other elements of the model by 100 mm in each of the X, Y, Z directions both with positive and negative sense (Figure 4—"D" mark). A definition of excessively small air space for strong source elements of the field leads to big computation errors due to artificial concentration of the magnetic field force lines.

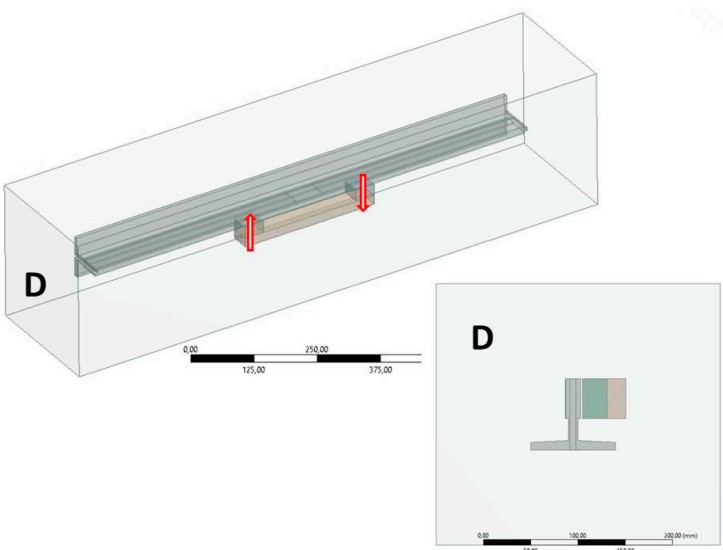

**Figure 4.** Model with air domain and polarization direction of permanent magnets.

## 5. Material Model

By defining the material model, first of all, the BH demagnetization curve was accepted for the permanent magnets made of "rare earth" materials. In this case, the NdFeB-N52 neodymium magnets with the following parameters were applied: residual flux density induction $B_r$ = 1450 mT, and coercive force $H_c$ = 800 kA/m. For the cramp, the curve of primary magnetization was defined starting the hysteresis loop of a magnetically soft material—steel SA1020 (Figure 5) and the guide rail—E235.

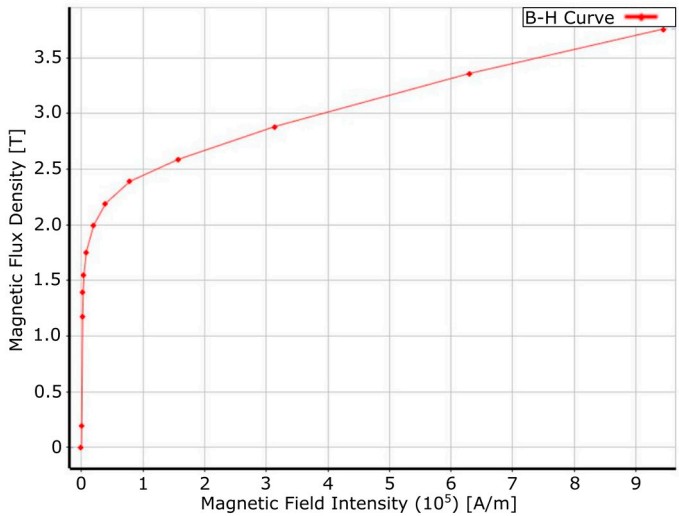

**Figure 5.** Curve of primary magnetisation of steel SA1020 used of the magnetic core cramp.

## 6. Boundary Conditions

The primary boundary condition in the magnetostatic analyses is to define the field input function, i.e., direction and sense of magnet polarization. In the case considered, the opposite senses of polarization were declared for both magnets so as to force a defined direction of the magnetic field flux flow within the entire circuit. The second and necessary boundary condition is to define the zero potential of the magnetic field on the boundary surfaces of the surrounding air domain (Figure 4).

## 7. Digitization

One of the most important stages of numerical analysis performance by means of the FEM is discretization of the analyzed model area (in case of magnetostatic analyses—also discretization of the air domain). The generated grid of finite elements should be adapted to the considered issue, considering both geometrical and physical aspects (Figure 6). Taking into account the sources of errors in numerical analysis and the need to minimize their impact onto the results of the obtained solutions, it is required to make several or more computation iterations, each time thickening the finite element grid [27]. This is the process of verifying the so-called convergence (concurrence) of the obtained results and making them independent from the finite element grid structure.

In the case discussed, the stray field area in the vicinity of boundaries (beginning and end of the modeled defect—Figure 6) was thickened by manual discretization and the appropriate tools of the ANSYS system related to global setting of the grid and local thickening (edges, planes). These are the areas where we particularly want to minimize the errors caused by multiple approximations. In the case of applying automatic discretization based on the existing adaptive algorithms, the model area is found where the accuracy of results should be increased and the FEM software makes decisions regarding the selection of areas and how to thicken the grid without possible interference of an analyst [28].

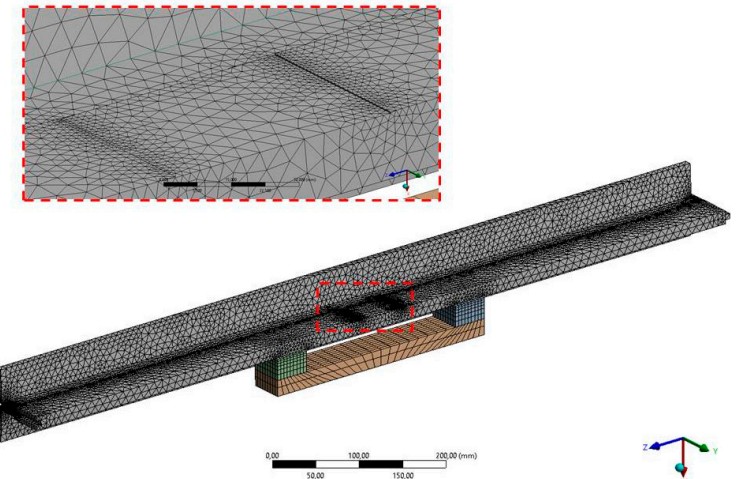

**Figure 6.** Model together with local thickening, without air domain.

The criterion for reaching the equilibrium (concurrence) [9] is the tolerance, which is expressed as a percentage. This is the reference to a difference between the magnetic induction values for which convergence is tested, which is obtained from the iterations occurring one by one. When each subsequent iteration generates smaller and smaller differences in values and such a difference is smaller than the established tolerance criterion, the further iterations are seized. The pursued value gets stabilized and is treated as an expected value, and the analysis is considered to be concurrent. A loose criteria of convergence (i.e., high tolerance of more than 10%) may lead to imprecise results, while a too low tolerance of convergence might cause an unjustified increase of "computation costs" (computation time, equipment requirements) needed to obtain the results to an "excessive" accuracy degree that does not add any significant changes to the value and the very analysis of results [12].

Apart from the very density of the grid, its quality parameters are essential as well, which depend on good approximations being obtained by using the shape function, especially in relation to the inter-node areas of a finite element. An exact structure of the finite element quality in the entire considered model is presented in Figure 7. The plot presents the percentage of finite elements within the entire model volume. The grid quality is assessed most often based on two typical quality parameters that can take different names in various numerical systems. In the ANSYS system, these are: Orthogonal Quality (OQ) and Skewness (SK). In the case of OQ, i.e., the geometrical regularity of

an element, it is required for a parameter to approach 1, while in the case of SK, i.e., the skewness of an element, it is required for a parameter to approach 0. In case of the OQ parameter and the tetra type elements, QO = 1 indicates a perfect tetrahedron, while for the hexa-type elements, QO = 1 indicates a perfect cube. Skewness takes the 0 value only for a perfect cube (elements of hexa type) as all angles are 90° and the element is not skew [20]. By analyzing the grid parameters during the analysis performed, the average value of Orthogonal Quality parameter was 0.852 (Figure 7) and the Skewness parameter was 0.2246. Based on both parameters, we can state that a very good quality of finite elements was obtained within the entire range analyzed. In the whole discretised d model, the dominating element is the tetra-type element, while the hexa-type elements constitute a marginal volume of the model.

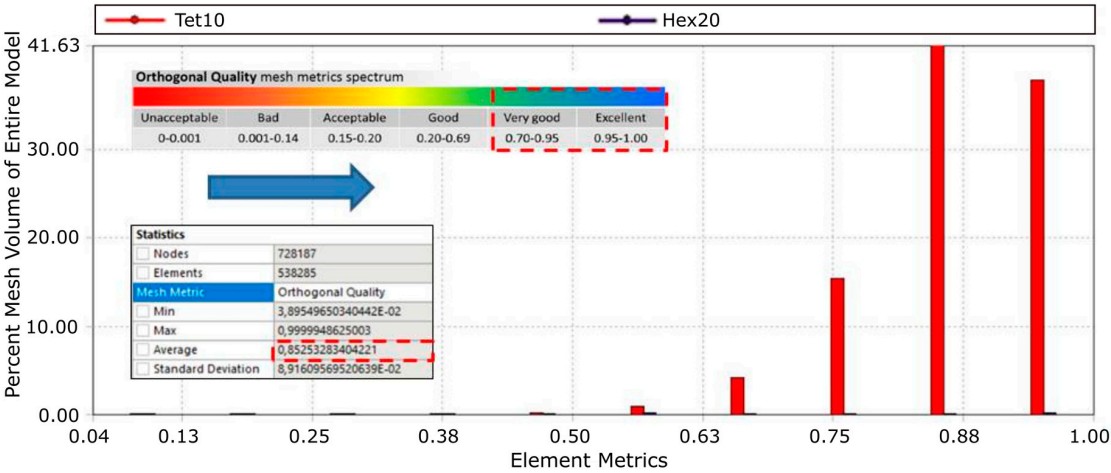

**Figure 7.** Structure of finite element grid in the model analyzed: orthogonal quality criterion.

## 8. Analysis of Results

The numerical analyses were performed to prove the capability of using the entire magnetic field and stray field to efficiently localize the guide rail damage locations after the operation of safety gears and interference of a maintenance technician (qualitative assessment), and to verify the capability of making quantitative assessment related to the guide rail wear degree versus the passenger lift service life. By analyzing the quality results by volume, for example, the contour plots of magnetic field induction in the entire model (magnetic core with air domain), it can be stated that the offered magnetic circuit with the determined dimensions and material parameters distributes the magnetic flux to the guide rail under consideration efficiently and at a sufficient level, which is proven by the induction value inside it; in the area between the magnets, it reaches approximately $B > 700$ mT, in the defect area, it reaches approximately $B = 1000$ mT, while the maximum value between the magnets reaches as much as approximately $B = 1600$ mT (Figure 8). Such a magnetization degree of the guide rail shall enable generating the stray field from the defect at a measurable level. A big value of magnetic energy is also proven by the high value of induction in the gap between magnets and the guide rail, which reaches even over $B = 1100$ mT (Figure 8).

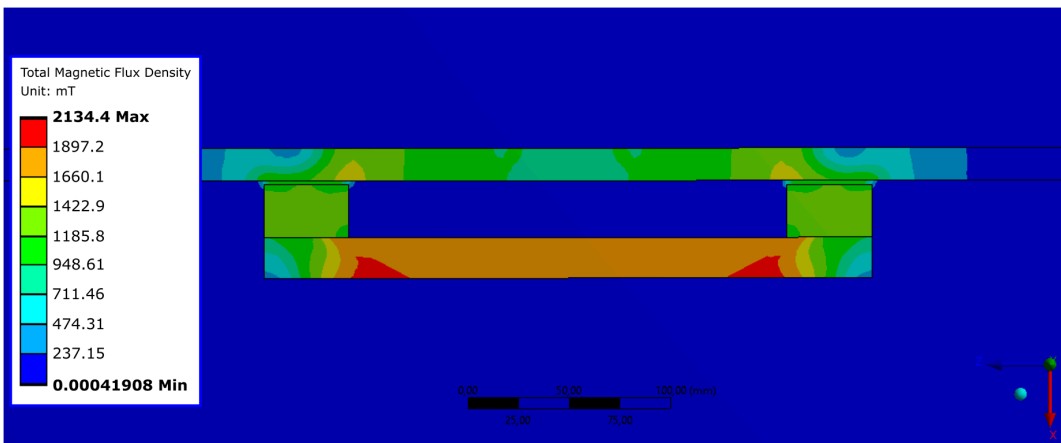

**Figure 8.** Field induction in magnetic core, guide rail, and air domain—cross-section.

Upon referring to the qualitative results concerning the magnetic field distributions on the surfaces of elements, it can be confirmed that the concept assumptions have been fulfilled. The distribution of magnetic induction vectors forming the magnetic field force lines over the guild rail surface as well as over the entire magnetic core confirms the enforced direction of magnetic flux flow and its looping through the guide rail (and specifically, by the guide rail section tested at a given time). In addition, the magnetic induction contour on the guide rail (Figure 9) indicates the magnetic field concentrations in an area between magnets, and in particular in the damage-affected and stray field magnetic detection zone.

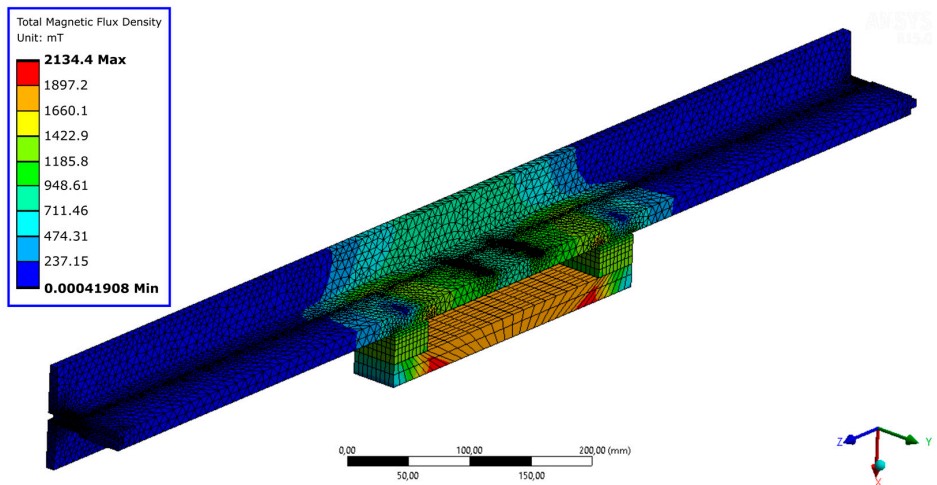

**Figure 9.** Distribution of magnetic field induction only in the guide rail and magnetic circuit of the head.

Referring specifically to the quantitative results related to the impact of wear size, in the analyzed range of variability of dimensions that characterize such wear, onto the parameters of stray field and magnetic induction in the very guide rail, several 2D slice and 3D plots were presented concerning the relevant correlation.

Figure 10 shows the 3D plot referring to the impact of size of the defect within:

- depth from 0.3 to 2.0 mm
- width from 50 to 150 mm

onto the maximum total magnetic induction in the guide rail, regardless of its location. It should be noted that globally, a change in the defect size within the range analyzed has a very negligible impact on the change in the value of maximum induction in the guide rail. The 3D plot surface curvature

(Figure 10) and its deformations may lead to different conclusions, but the range of changes in value corresponding to such a curvature is from approximately 1618 to 1630 mT.

　　The analogical plot is presented in Figure 11, but it refers to the data collected from a different area. Such an area was the previously defined detection line, which is called the "path". This line limits the trajectory of a sensor that measures the magnetic field (Figure 12). It should be only noted that in the target solution, the sensor shall move together with the magnetic core along the head. In this numerical analysis, we consider the magnetostatic case where the guide rail, the head, and the field generated remain at rest.

　　Implicitly, the sensor indicating the magnetic field value is shifted along the detection path. The maximum possible number of 200 test points spaced evenly was defined on the detection path (it concern FEM analysis). The defined "path" passes through the center of the track of the guide rail, taking into account two planes of symmetry (longitudinal and lateral). Thus, it passes also through the center of the modeled wear. Taking into account the limitations of this sensor, the detection path is shifted from the original surface of one guide rail by 0.3 mm. The beginning and the end of the line are located each time 15 mm away from the guide rail wear edge in the outer direction, and this location was updated during computations of subsequent iterations related to a change of the wear size (Figure 12).

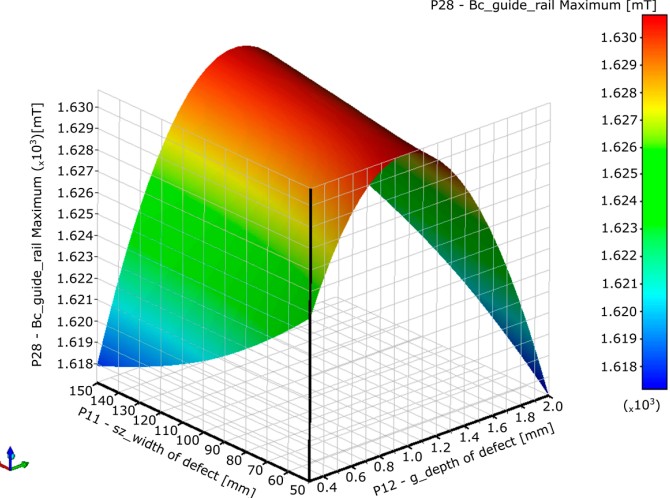

**Figure 10.** 3D plot showing the impact of width "*sz*" and depth "*g*" of defects onto the maximum value of total induction in the guide rail.

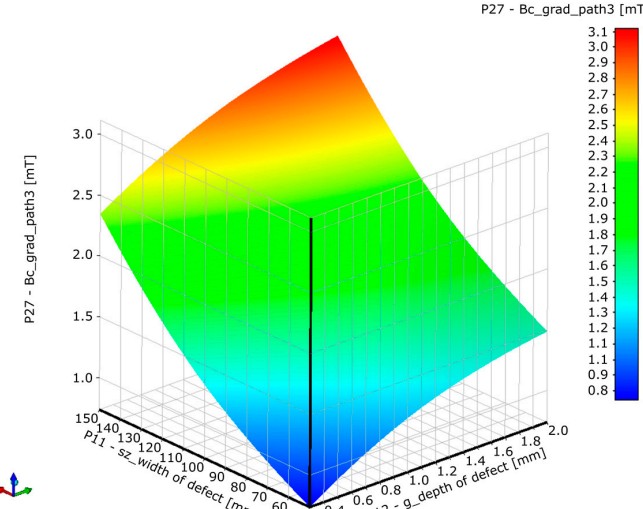

**Figure 11.** 3D plot showing the impact of width "sz" and depth "g" of defect onto a gradient of magnetic induction on the detection path.

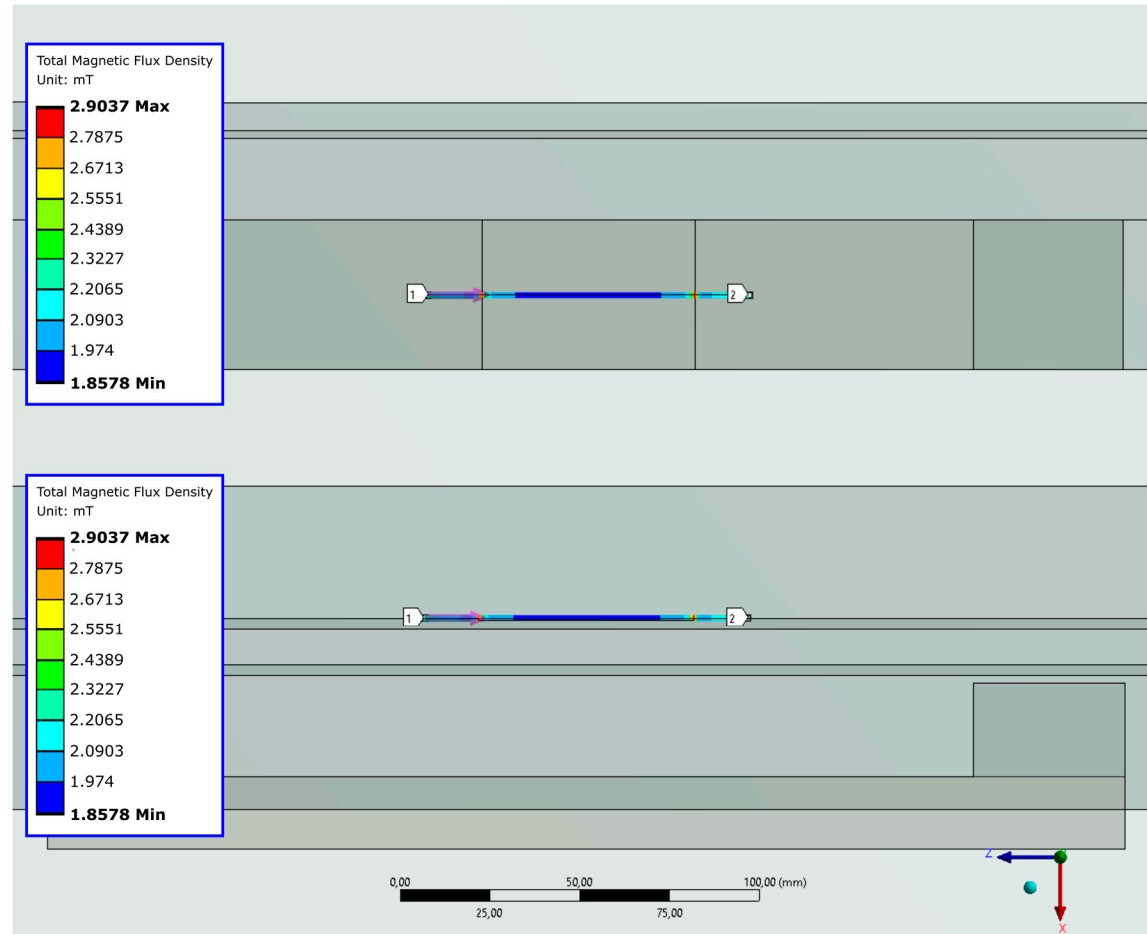

**Figure 12.** Form of detection line (called "path").

The values being maximum and minimum ones are selected from the data collected along the detection path from 200 test points, and the gradient of these values is computed. Figure 11 presents the impact of the defect dimensions *g* and *sz* onto the value of the gradient from the detection path. This gradient ranges from approximately 0.8 to 3.1 mT, which does not constitute any detection problem, taking into consideration the common availability of the 1 nT-sensitivity sensors on the market. There is a clear tendency that both the increase of wear depth and its width along the guide rail result in higher gradient values that have a positive effect on the detectability, which is logical and desirable. Here, the parameter of defect depth rather than defect width has a bigger impact onto the gradient. This is also confirmed by the 2D slice plot in form of nomograms presenting a gradient from the detection path on the vertical axis and one of the defect parameters changing in a continuous manner on the horizontal axis, while the colorful curves are the second parameter of defect that change by increments (Figure 13).

The nomograms of this type, when recomputed into actual measurements and upon calibration of the head on the models of worn out guide rails, could be used as the calibration curves for reading out the wear based on the diagnostic signal from the sensor or as a tool for automating the wear degree computation process.

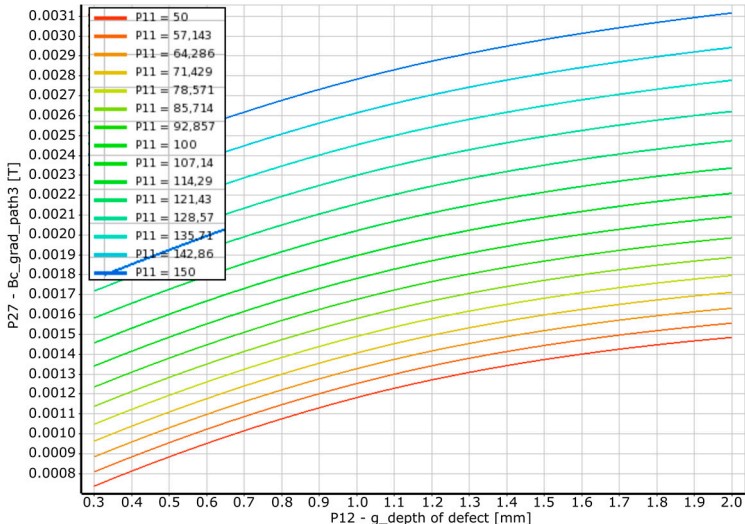

**Figure 13.** Gradient $B_c$ from detection path versus continuous change of defect width and step change of defect depth.

Summing up, the final numerical experiment with the use of FEM in magnetostatic analysis confirmed the concept defined in the patent. Generally, the analysis explicitly confirmed that the assumptions were correct and that the stray field (field disturbance) can be used to identify the locations and define the wear of guide rails. The accomplished analysis proved, regarding the qualitative assessment, that the gradients (magnetic field concentration locations) form at the boundary of notches generated after grinding, which enables detecting them using appropriate induction, Hall-effect, or magnetometric sensors. Regarding the qualitative assessment, taking into account the field gradient value in correlation with the defect size (Figure 13) and knowing the length of a defect, we can determine its depth. For example, if the velocity of the head movement in relation to the guide rail is $V = 100$ mm/s, while the interval between the recorded local field gradients is $t = 1$ s, then the defect width is $sz = 100$ mm. If the amplitude of such gradients reaches approximately $B_c = 1.5$ mT, then the depth of such wear determined on the basis of the mentioned plots is approximately $g = 0.51 \div 0.52$ mm. Should there be a need to increase the stray field change amplitude, the use of a bigger number of magnets may be considered (bigger volume of magnetic energy), but the increase of cohesion forces between the ferromagnetic guide rail and the flux-distributing magnet or pole shoe, especially at such small (3 mm) air gaps must be taken into account as well.

## 9. Conclusions

On the basis of the numerical analyses carried out, it was established that:

- The final evidence in the form of changes of the induction components in stray magnetic field $B_x$, $B_z$, and $B_c$ collected from 200 test points on the detection path are presented in Figure 14. The component $B_y$ was ignored intentionally, as it did not show any variability tendency or level that would be significant for the scale of the problem.
- The identified level of the stray field variability resulting from defects makes the detection definitely possible for the current measurement technology.
- The many years of experience forming the magnetic circuits and the accomplished analysis proved that it was correct to assume that the sensor should be located on the opposite side of the magnetic core. The experiments prove that the presence of a cramp at a close distance from the tested component (guide rail) sometimes results in capturing the magnetic field directly by air (beyond the circuit), which results in additional disturbance in the operation of the sensor if it is located on the side of magnetic core (within its range).

- It was also proved that the defect size had no significant impact, within the range tested, onto the changes in the maximum value of magnetic induction inside the guide rail.

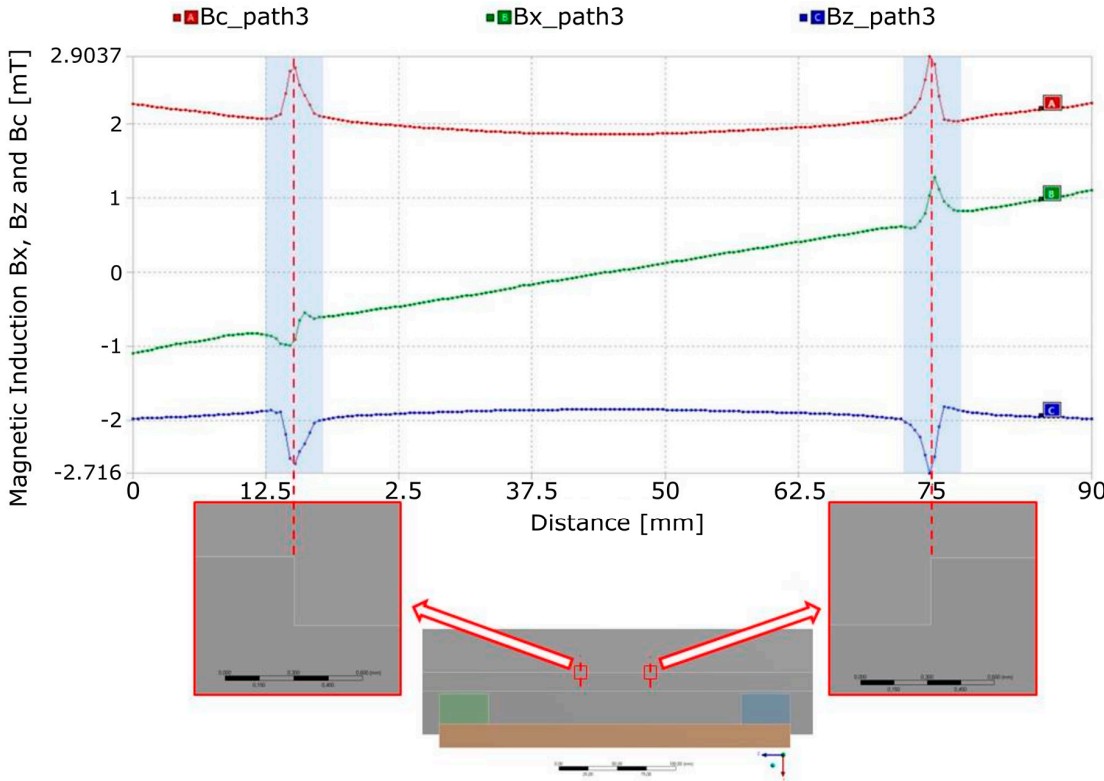

**Figure 14.** Magnetic field along the detection path and disturbance of such components as stray field representatives in the wear zone.

**Author Contributions:** All authors contributed to the study conception and design; methodology: P.L., H.R.; formal analysis and investigation: P.L., H.R., T.K.; writing—original draft preparation: P.L., H.R., T.K.; writing—review and editing: P.L., H.R.; funding acquisition: P.L., H.R., T.K. All authors have read and agreed to the published version of the manuscript.

**Funding:** The studies were not introduced under any financial support.

**Conflicts of Interest:** The authors declare that there is no conflict of interest regarding the publication of this paper.

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
