# Peer review of "Application of Stray Magnetic Field for Monitoring the Wear Degree in Steel Components of the Lift Guide Rail System"

_metals, doi:10.3390/met10081008_

Round 1
Reviewer 1 Report
The paper presents an interesting way to make wear on guide rails for elevators measurable. I am concerned about the journal choice, at it has nothing to do with “metals”, it is more suitable for a sensor journal.
It presents a lot of graphs taken from the same simulation, which should be substantially reduced, as many are redundant or unnecessary (like the mesh quality). Further it lacks a discussion part, which is essential to understand the impact of the work onto sensor design.
I also have the impression, that the paper should advertise the authors patent – surely it should be cited, but nothing more is relevant for a scientific paper.
The English language is poor and needs proof-reading
Abstract: patent reference unnecessary here
Line 112-118: why do the references jump from 10 to 36?
A clear aim of this study should be stated at a reasonable position, e.g. at end of chapter 1 or 2.
Line 132,186: reference is missing
Fig. 2a: this is a drawing
Line 153: a table would be better
Is the paper’s author and the patent owner the same person? Reference to the patent is missing, Patent-Number is wrong, it should be PL423758
Line 204: I doubt 10 references to the equations are necessary
Eq. 1: J is missing
Nable, rot, div do not need explanation
Formula symbols should be italic, also in the text
Line 290: paragraph is off
A space between value an unit is necessary every time!
Fig. 5: can you add the formula symbols to the axis; no subscript x, superscript -1
Line 323: digitization
There is no such thing as mesh “thickening”, its refinement
I think we can trust you on net quality, no need to highlight it here in great detail
Fig 11 ff: the placement of the legend is suboptimal, there is bad contrast in some cases, and often there is an unfitting background in the legend. Make sure that is clearly readably like in Fig. 14.
Fig. 17/18, 20/21 are redundant, chose one depiction.
Fig. 11ff: there are many depictions of the same results, please reduce it to the absolute minimum. A usual paper has max. 10-12 figures
Line 451: limitations
A discussion of the impact of the work on elevators, maintenance, safety should be given. Also the benefits and drawbacks to other measurement techniques (like optical distance sensors) shall be given.
The conclusions are mainly what I would consider a discussion. In the conclusions no new aspects, data or implications should be discussed. Also new images or references to prior images have no place in the conclusions. It should only summary the main results of the work!
References:
Patent is missing. Also it is not accessible by standard patent search engines, so it is difficult to judge
11 self-citations of the authors are clearly too much in a single paper
16 articles are polish and hence not readable by the majority of the readers – could you find adequate English publications or skip them?
Author Response
Dear Reviewer
I'd like thanks you for sent full report for our manuscript. I attached our response to the yours raports.
Best regards
Authors

Reviewer 2 Report
The objective of this paper was clearly presented. The whole paper was organized well. I suggest accepting the paper in the current formation.
Author Response
Dear Reviewer
I'd like thanks you for sent full report for our manuscript. I attached our response to the yours raports
Reviewer 3 Report
- At the end of introduction, I suggest that you add a few comments describing what are the differences of this work with the literature that you are describing previously. That will clearly show the added value of this manuscript.
- Some references are not displayed correctly (e.g. line 132).
- There are also some grammatical mistakes within the text (e.g. line ‘imitations’ should be ‘limitations’).
- From Figure 25, indeed one can see that the magnetic induction changes with distance. These changes are indicated where you have a sharp change in the surface topography. However, wear is quite a complicated and dynamic phenomenon, resulting in complex wear track morphology. For example, what will happen if you have material transferring (adhesive wear), oxidative wear or a combination of different mechanisms (adhesion and abrasion)? To you expect to see a similar behavior.
- What is the resolution of this method?
- This is an interesting work, but my main concern is how it relates to the actual wear of the component. Maybe you should add some comments to describe how this can be achieved.
Author Response

(The authors gave the same response as above.)

Round 2
Reviewer 1 Report
Line 144: grammar needs improvement
Table requires table caption
You stated in the answers to the reviewer’s comments, that the author of the paper and the patent are the same, so why is it Poul Lonkwic for the paper and Paweł Lonkwic for the patent?
Reference 37 does not work
Line 194: double naming of reference 23
Between Fig. and number often the space is missing
Line 446-447: grammar needs improvement
There is still a discussion of possible other measurement methods missing. Optical sensors, eddy current sensors, etc. will provide much more accurate results, so what is the benefit of your method? A critical discussion is mandatory!
Line 446: Italic “By”
Author Response
Dear Reviewer
I'd like to thank you for sending the full report referring to our manuscript.
Below please find the short answers to your suggestions:
- Line 144: grammar needs improvement
Ans.: The grammar has been changed.
- Table requires table caption
Ans.: The table caption has been added.
- You stated in the answers to the reviewer’s comments, that the author of the paper and the patent are the same, so why is it Poul Lonkwic for the paper and Paweł Lonkwic for the patent?
Ans.: The author's name has been standardized.
- Reference 37 does not work
Ans.: Reference 37 has been removed from the text.
- Line 194: double naming of reference 23
Ans.: The double naming of reference has been deleted.
- Between Fig. and number often the space is missing
Ans.: The spaces have been added.
- Line 446-447: grammar needs improvement
Ans.: The grammar has been corrected.
- There is still a discussion of possible other measurement methods missing. Optical sensors, eddy current sensors, etc. will provide much more accurate results, so what is the benefit of your method? A critical discussion is mandatory!
Ans.: The discussion has been added in the lines 160-178.
- Line 446: Italic “By”
Ans.: “By” has been changed to italics in the line 470.
Best regards,
Poul Lonkwic